# Polarization Modulation Instability in Dispersion-Engineered Photonic Crystal Fibers

**Abraham Loredo-Trejo** [1,2], **Antonio Díez** [1,2,*], **Enrique Silvestre** [1,3] and **Miguel V. Andrés** [1,2]

1 Laboratory of Fiber Optics, ICMUV, Universidad de Valencia, 46100 Valencia, Spain; abraham.loredo@uv.es (A.L.-T.); enrique.silvestre@uv.es (E.S.); miguel.andres@uv.es (M.V.A.)
2 Departamento de Física Aplicada y Electromagnetismo, Universidad de Valencia, 46100 Valencia, Spain
3 Departamento de Óptica, Universidad de Valencia, 46100 Valencia, Spain
* Correspondence: antonio.diez@uv.es

**Abstract:** Generation of widely spaced polarization modulation instability (PMI) sidebands in a wide collection of photonic crystal fibers (PCF), including liquid-filled PCFs, is reported. The contribution of chromatic dispersion and birefringence to the net linear phase mismatch of PMI is investigated in all-normal dispersion PCFs and in PCFs with one (or two) zero dispersion wavelengths. Large frequency shift sidebands are demonstrated experimentally. Suitable fabrication parameters for air-filled and liquid-filled PCFs are proposed as guidelines for the development of dual-wavelength light sources based on PMI.

**Keywords:** photonic crystal fiber; polarization modulation instability; ANDi fiber; liquid-filled PCF





## 1. Introduction

In the last decades, great effort has been applied to take advantage of nonlinearity in optical fibers for the development of new light sources as supercontinuum sources [1,2] or four-wave mixing (FWM)-based light sources. Particularly, solid-core photonic crystal fibers (PCF) [3] are an excellent platform for the generation of nonlinear processes, owing to the large flexibility for fiber chromatic dispersion design, the strong optical density, and the large interaction lengths available. FWM has attractive applications in different areas of interest: parametric amplifiers [4], frequency combs based on cascade-FWM [5], dual-wavelength light sources for special microscopy as coherent anti-Stokes Raman spectroscopy [6,7], and novel nonlinear quantum applications as photon-pair sources [8]. FWM is a flexible, nonlinear parametric process in which two pump photons of different (or identical) frequency interact with a nonlinear material to produce two photons frequency-shifted from the pump, one with higher energy than the pump, called anti-Stokes photon, and another with lower energy, called Stokes photon. The frequency shift obeys the energy conservation relation and phase matching condition [9].

In the most common cases, FWM applications rely on the scalar FWM process, in which the state of polarization of the involved photons is identical. Light sources with specific characteristics of polarization can be of great interest in the area of optics. In this sense, vector–FWM (V–FWM) can be exploited for the development of such light sources. V–FWM is ruled by the interplay of the state of polarization of the involved photons with the nonlinear medium. We find two different types of V–FWM processes, named polarization modulation instability (PMI) and cross-phase modulation instability (XPMI) [10]. XPMI can be generated in fibers with moderated birefringence. Linearly polarized light oriented at 45°, with respect the principal fiber axes, produces two sidebands with orthogonal polarization. PMI is mostly generated in very weakly birefringent fibers, in which a linearly polarized optical pump signal with polarization oriented along one fiber axis produces two sidebands with the same polarization state but orthogonal, with respect to the polarization of the pump beam. PMI generation was extensively studied

and experimentally demonstrated in standard optical fibers in the past [11–13]. In PCFs, PMI was experimentally demonstrated for the first time by Kruhlak et al. [14] in a PCF with normal dispersion at the pump wavelength and pumping near the zero dispersion wavelength (ZDW). In recent works, we have reported experimental PMI generation in all-normal dispersion (ANDi) fibers [15].

Filling the holes with optical materials has been demonstrated to be a useful postfabrication technique for changing the dispersion properties of PCFs. Additionally, it enables fine tuning of the chromatic dispersion, by using the sensitivity of the filling material with external parameters, as, for example, temperature. Several applications related to the nonlinear properties of liquid-filled PCFs have been reported [16–18]. In particular, we demonstrated wide frequency tuning of strong PMI sidebands in ethanol-filled PCFs [19].

Here, we report theoretical simulations and a full experimental study of PMI effect, investigated in a collection of air-filled, ethanol-filled, and heavy water ($D_2O$)-filled PCFs with different structural parameters and chromatic dispersion characteristics. We used the energy conservation and phase matching condition relations to explore the dispersion and birefringence contributions to PMI frequency shift in ANDi fibers and in fibers with one or two ZDWs.

## 2. Theoretical Modelling

In the context of optical fibers, PMI is a nonlinear parametric process, in which two photons with linear polarization oriented to one of the principal fiber axes are annihilated to give rise to two new photons with different frequencies, same polarization state, but orthogonal, with respect to pump photons. PMI is generated in singlemode optical fibers through coherent coupling of the two polarized eigenmodes $HE_{11x}$ and $HE_{11y}$, whose propagation factors are slightly different due to unintentional residual linear birefringence present in the fiber. Depending on the polarization of pump photons, two PMI processes can develop. In the first scenario, the polarization of pump photons is aligned with the slow axis of the fiber, and the PMI photons are generated with polarization oriented to the fast axis. This process is called PMI–SF. In the opposite case, PMI–FS, pump photons are polarized along the fast axes, and the generated photons are polarized along the slow axes.

PMI–SF can produce two sidebands widely spaced and detuned from the pump, even with low pump power. In contrast, PMI–FS presents a power threshold $P_{th} = 3 \cdot \Delta n \cdot A_{eff} / 2 \cdot n_2$ that needs to be reached before PMI generation, where $\Delta n$ is the modal birefringence, $A_{eff}$ is the modal effective area and $n_2$ is the fiber nonlinear refractive index. PMI–FS features are somehow more complex, due to the pump power requirements. If $P < P_{th}$, PMI is not generated. When $P_{th} < P < 2P_{th}$, PMI gain shows a maximum value at the zero detuning with no sidebands generation. Only with $P > 2P_{th}$ are sidebands generated close to pump, but with gain remaining different from zero at zero detuning.

Spectral position of PMI sidebands can be calculated from the energy conservation and net phase mismatch relation. The phase–matching condition for both PMI process are described by the following phase equation [9],

$$2\beta_P - \beta_{AS} - \beta_S \pm \frac{\Delta n \times (\omega_{AS} + \omega_S)}{c} + \frac{2}{3}\gamma P = 0 \qquad (1)$$

where $\beta_P$, $\beta_{AS}$ and $\beta_S$ are the propagation factors of the fundamental mode at the pump wavelength, anti-Stokes and Stokes wavelengths, respectively; $\omega_{AS}$ and $\omega_S$ are the frequencies of the generated photons; $c$ is the speed of light in the vacuum; $P$ is the pump power and $\gamma$ is the nonlinear coefficient, defined as $\gamma = n_2 k_p / A_{eff}$, where $k_p$ is the pump wavenumber. In Equation (1) the upper sign stands for PMI–SF, while the lower sign stands for PMI–FS. It can be seen in Equation (1) that both chromatic dispersion and birefringence contribute to the linear phase term. Therefore, PMI frequency shift is strongly affected by both fiber parameters.

We study theoretically the generation of PMI–SF in solid-core PCF with holes filled with air, with ethanol and with heavy water. Figure 1 shows a simple scheme of a solid-core

PCF cross-section showing the structural parameters (i.e., hole diameter *d* and pitch $\Lambda$). First, the linear properties of the fibers were calculated using a fully-vector method [20]. Then, Stokes and anti-Stokes wavelengths were obtained from Equation (1) and the energy conservation condition. Simulations of the PMI–FS process are not included, because it was not investigated in the experiments due to the existence of a power threshold.

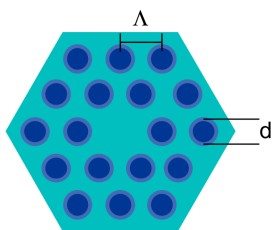

**Figure 1.** Scheme of cross-section of solid-core photonic crystal fibers (PCF).

The chromatic dispersion of air-filled PCF can be tailored just by changing the fiber structural parameters. Several combinations of *d* and $\Lambda$ can be suitable/nm for attaining given dispersion requirements. Figure 2a shows the theoretical calculation of chromatic dispersion for air-filled PCF as a function of the optical wavelength and the fiber pitch. A fixed value of $d/\Lambda = 0.36$ was used in these simulations. The blue level curve in Figure 2a indicates zero dispersion. For this specific air-filling fraction, we can distinguish two different dispersion regimes: (i) when $\Lambda < 1.8$ μm the dispersion presents an ANDi profile, and (ii) when $\Lambda > 1.8$ μm the dispersion profile shows at least one ZDW. In the first regime, fibers with $\Lambda$ approaching 1.8 μm show low dispersion values in the wavelength range under study, increasing significantly as $\Lambda$ becomes smaller.

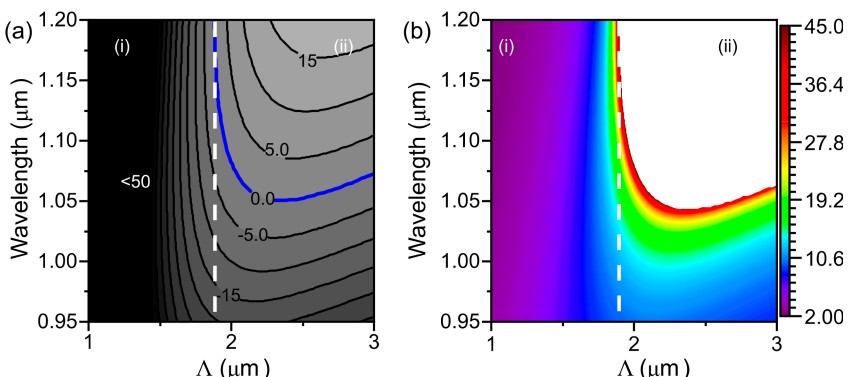

**Figure 2.** (**a**) Chromatic dispersion as a function of optical wavelength and fiber pitch for air-filled PCFs. Numbers indicates dispersion levels in ps/km·nm. White dashed line indicates pitch of 1.8 μm. (**b**) Frequency shift map of polarization modulation instability (PMI–SF); color bars are detuning frequencies values in THz. Parameters used in the simulations: $d/\Lambda = 0.36$, *P* = 3 kW.

Figure 2b shows a frequency shift map of the PMI–SF process as a function of the optical pump wavelength and fiber pitch. Additional parameters used in PMI–SF simulation are the following: pump power of 3 kW, phase birefringence $\Delta n = 1 \times 10^{-6}$ and nonlinear refractive index $n_2 = 2.7 \times 10^{-20}$ m²/W. In the case of ANDi fibers, i.e., dispersion region (i), PMI–SF is generated in all cases. For low $\Lambda$ values, the frequency shift is relatively small, and it shows little dependence on the pump wavelength. The frequency shift increases as the fiber pitch increases, and/or the dispersion at the pump wavelength decreases. In dispersion region (ii), the result is more complex. For $\Lambda$ slightly above 1.8 μm and pump wavelengths in which the dispersion is small and near to zero (pump wavelength approaching ZDW), PMI–SF experiences a large frequency shift. Additionally, there is a forbidden region where no phase matching solutions for PMI–SF can exist. PMI generated

in fibers with one ZDW present a spectral turning point, beyond which the PMI phase matching condition is no longer fulfilled. This occurs when the dispersion mismatch cannot compensate for the contribution of nonlinear mismatch and residual birefringence of the fiber.

The dispersion properties change after filling the holes of PCFs with material different from air, due to the increase of refractive index in the fiber microstructure and the dispersion properties of the new material. For our experiments, we chose ethanol and heavy water, due their suitable refractive index and low absorption in the infrared region. Refractive index characteristics of both liquids used for the theoretical simulations are taken from [21]. Their wavelength dispersion was taken into account in the calculations Figures 3a and 4a show dispersions maps calculated for fibers filled with ethanol and $D_2O$, respectively. A value of $d/\Lambda = 0.56$ was chosen for $D_2O$-filled fibers, and $d/\Lambda = 0.58$ for ethanol-filled fibers. These parameters correspond to the mean values of structural parameters of fibers used in the experiments. In both cases, the zero dispersion level occurs for larger wavelengths and larger pitch lengths, compared to air-filled fibers. Figures 3b and 4b show the corresponding frequency shift maps for PMI–SF produced in ethanol-filled and $D_2O$-filled fibers, respectively. As in the previous case, larger frequency shift occurs in fibers that exhibit low dispersion at the chosen pump wavelength. The shift of zero dispersion that ethanol-filled and $D_2O$-filled fibers exhibit also causes the movement of the forbidden region of PMI to longer wavelengths and larger pitch lengths.

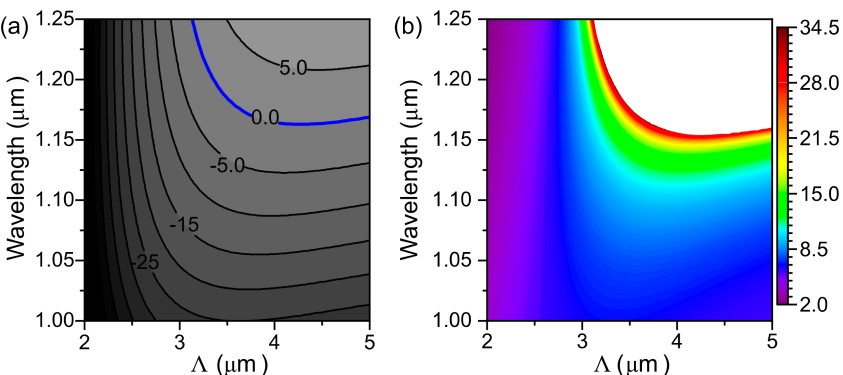

**Figure 3.** (**a**) Chromatic dispersion as a function of optical wavelength and fiber pitch for ethanol-filled PCFs. Numbers indicates dispersion levels in ps/km·nm. (**b**) Frequency shift map of PMI–SF; color bars are detuning frequencies values in THz. Parameters used in the simulations: $d/\Lambda = 0.58$, $P = 3$ kW.

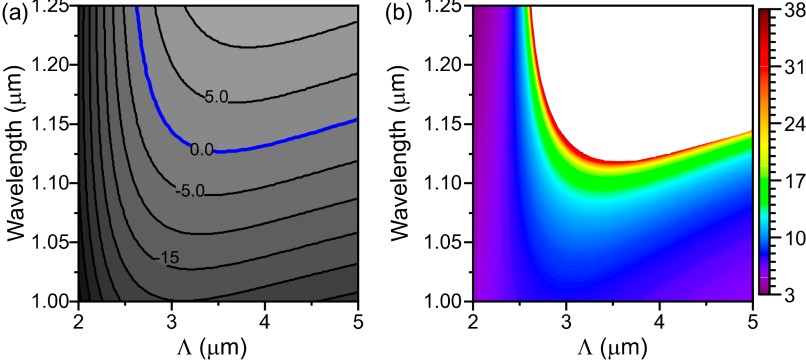

**Figure 4.** (**a**) Chromatic dispersion as a function of optical wavelength and fiber pitch for $D_2O$-filled PCFs. Numbers indicates dispersion levels in ps/km·nm. (**b**) Frequency shift map of PMI–SF; color bars are detuning frequencies values in THz. Parameters used in the simulations: $d/\Lambda = 0.56$, $P = 3$ kW.

## 3. Characteristics of the PCFs

We carried out several experiments to study PMI generation with sub-nanosecond pump pulses in conventional PCFs with air holes and in PCFs filled with different optical liquids. A collection of fibers was made of silica, following the stack-and-draw technique. The PCFs comprise a microstructure with triangular lattice of air holes with a missing hole at the center of the microstructure that acts as the fiber core. After the fibers were fabricated, some of them were infiltrated with ethanol or with $D_2O$ through capillarity force and gas pressure. All fibers were singlemode at the experimental wavelength range. Physical and waveguiding characteristics of the fibers are described in the following sections.

### 3.1. Air-Filled PCFs

Air-filled PCFs were designed and fabricated to present a convex-ANDi profile in a broad wavelength range. Figure 5a shows scanning electronic microscope (SEM) images of the cross section of the fibers used in the experiments. The structural parameters of the fibers were obtained from the SEM images and are summarized in Table 1. Chromatic dispersion characteristics and effective area were calculated by the method described in [20]. Figure 5b shows the chromatic dispersion as function of wavelength of the four PCFs. Table 1 also includes dispersion values and effective area of the fundamental mode at 1064 nm (i.e., the experimental pump wavelength). In all fibers, the maximum of the curve, i.e., the minimum dispersion wavelength (MDW), is close to the pump wavelength and has a relatively small dispersion value, at 1064 nm.

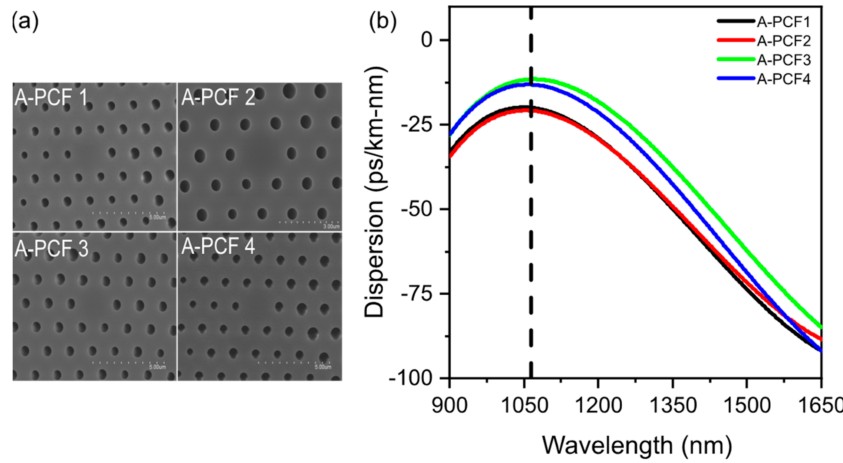

**Figure 5.** (**a**) SEM images of air-filled PCFs cross-section. (**b**) Chromatic dispersion of air-filled PCFs. Dashed vertical black line indicates the experimental pump wavelength.

**Table 1.** Structural parameters, dispersion and effective mode area at 1064 nm of air-filled PCFs.

| Fiber | $\Lambda$ (µm) | $d/\Lambda$ | Dispersion (ps/km·nm) | $A_{eff}$ (µm²) |
|---|---|---|---|---|
| A-PCF1 | 1.57 | 0.35 | −19.9 | 7.2 |
| A-PCF2 | 1.59 | 0.35 | −20.8 | 7.5 |
| A-PCF3 | 1.60 | 0.36 | −11.5 | 6.7 |
| A-PCF4 | 1.57 | 0.37 | −13.1 | 6.5 |

### 3.2. Ethanol-Filled PCF

A collection of PCFs with the appropriate dispersion characteristics for PMI generation with the microstructure holes filled with ethanol were designed and fabricated. Figure 6a shows their SEM images taken before infiltration, and Table 2 gives the structural parameters of the fibers. The guiding characteristics were calculated, taking into account the refractive index of ethanol, including its wavelength dispersion [21]. Figure 6b shows

the dispersion profile of the six fibers investigated. All of them exhibit normal dispersion at 1064 nm. According with their dispersion characteristics, different types of fiber were obtained. Three fibers showed an ANDi profile (E-PCF1, E-PCF2 and E-PCF3), with different values of dispersion at the MDW and increasing values of dispersion at 1064 nm. Two fibers (E-PCF4, E-PCF5) showed a dispersion profile that shows two ZDWs, the first near to 1200 nm and second at 1600 nm, quite far from the pump wavelength. These two fibers have very similar characteristics, although not identical. The results of both PCFs have been included to show that PMI frequency shift can be very sensitive to the fiber properties. Finally, we included in this study one PCF (E-PCF6) with a dispersion profile showing just one ZDW ($\approx$1117 nm) in the vicinity of the pump wavelength and a small value of normal dispersion at the pump wavelength. It is noteworthy that the chromatic dispersion of all these six fibers having air in the holes are anomalous at the pump wavelength, with the ZDW quite below the pump wavelength.

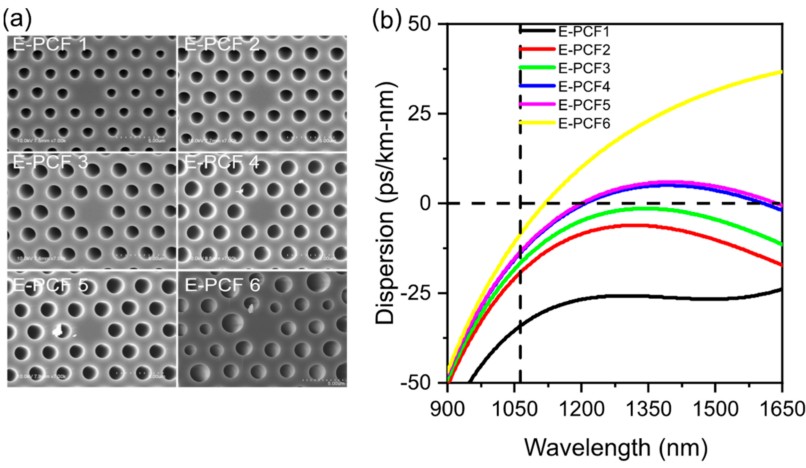

**Figure 6.** (**a**) SEM images of ethanol-filled PCFs cross-section. (**b**) Chromatic dispersion of PCFs with holes filled ethanol. Dashed vertical black line indicates the experimental pump wavelength.

**Table 2.** Structural parameters, dispersion and effective mode area at 1064 nm of ethanol-filled PCFs.

| Fiber | $\Lambda$ ($\mu$m) | $d/\Lambda$ | Dispersion (ps/km·nm) | ZDW (nm) | $A_{eff}$ ($\mu$m$^2$) |
|---|---|---|---|---|---|
| E-PCF1 | 2.66 | 0.43 | −34.1 | - | 20.1 |
| E-PCF2 | 2.76 | 0.54 | −19.1 | - | 12.3 |
| E-PCF3 | 2.83 | 0.57 | −16.1 | - | 11.6 |
| E-PCF4 | 2.90 | 0.59 | −13.7 | 1208 | 11.2 |
| E-PCF5 | 2.92 | 0.60 | −13.4 | 1200 | 11.2 |
| E-PCF6 | 3.97 | 0.75 | −8.4 | 1117 | 13.1 |

### 3.3. D$_2$O-Filled PCF

Figure 7a shows SEM images of the fibers that were infiltrated with D$_2$O. Table 3 gives their structural parameters as well as the chromatic dispersion and effective area at 1064 nm. The guiding properties were calculated, taking into account the refractive index of D$_2$O and its wavelength dispersion [21]. Figure 7b shows their dispersion characteristics. All of them exhibited normal dispersion at pump wavelength. One PCF (D-PCF1) showed an ANDi profile, and the dispersion profile of the rest of PCFs showed one ZDW at different values of wavelength, from 1092 nm to 1140 nm, and increasing dispersion values at 1064 nm, from −9.1 ps/nm km to −3.9 ps/nm km.

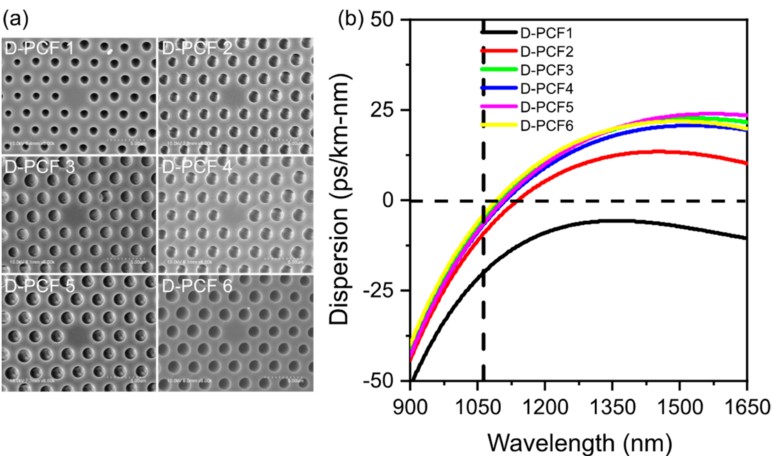

**Figure 7.** (**a**) SEM images of $D_2O$-filled PCFs cross-section. (**b**) Chromatic dispersion of PCFs with holes filled with $D_2O$. Dashed vertical black line indicates the experimental pump wavelength.

**Table 3.** Structural parameters, dispersion and effective mode area at 1064 nm of $D_2O$-filled PCFs.

| Fiber | $\Lambda$ (µm) | $d/\Lambda$ | Dispersion (ps/km·nm) | ZDW (nm) | $A_{eff}$ (µm²) |
|---|---|---|---|---|---|
| D-PCF1 | 2.68 | 0.44 | −19.9 | - | 15.0 |
| D-PCF2 | 2.78 | 0.54 | −9.1 | 1140 | 10.9 |
| D-PCF3 | 2.86 | 0.59 | −5.3 | 1102 | 9.9 |
| D-PCF4 | 2.92 | 0.56 | −6.6 | 1112 | 11.2 |
| D-PCF5 | 2.95 | 0.58 | −6.2 | 1107 | 10.8 |
| D-PCF6 | 2.76 | 0.62 | −3.9 | 1091 | 9.0 |

## 4. Experimental Setup

Figure 8 shows the experimental arrangement scheme. The pump laser is a passively Q-switched Nd: YAG microchip laser (TEEM Photonics SNP 20F-100) that emits pulses at 1064.5 nm of 700 ps duration (FWHM), few kW of peak power, and a repetition rate of 19.1 kHz. The laser emission is linearly polarized with a polarization extinction ratio (PER) of 32 dB. The laser beam was delivered into the fiber under test (FUT). The length of the fibers used in the experiments was about ≈1 m. A half-wave plate (HWP) was used to rotate the polarization plane of the pump beam. The light exiting the optical fiber was collected and analyzed with an optical spectrum analyzer (OSA).

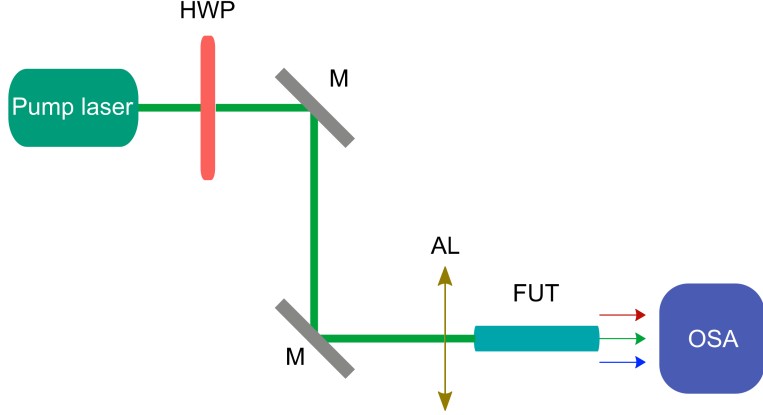

**Figure 8.** Experimental arrangement. Half-wave plate (HWP). Mirror (M). Aspherical lens (AL). Fiber under test (FUT). Optical spectrum analyzer (OSA).

## 5. Experimental Results and Discussion

Light spectra from the fibers' output were recorded when the polarization orientation of the pump was aligned to the main fiber axes. Figure 9 shows the spectrum obtained from fiber A-PCF1 for two orthogonal polarization orientations and the same pump power. The black line corresponds to pump polarization, matching the slow axis of the fiber. Two strong sidebands, centered at 991 nm and 1148 nm, are generated in the fiber. These two sidebands refer to anti-Stokes and Stokes photons generated via the PMI–SF process. The blue line shows the spectrum obtained after the pump polarization orientation was rotated by 90° (i.e., fast-axis pumping). The strong sidebands described above are not present anymore, and the rest of the spectrum remains practically the same. PMI–FS was not observed in the experiment, as the pump power level ($\approx$2.4 kW) injected into the fiber was below the power threshold required for PMI–FS to occur in this fiber ($P_{th}$ = 6.4 kW). The two secondary sidebands of lower amplitude, centered at 1016 nm and 1116 nm (frequency shift $\approx$13.2 THz) and shown in both spectra, are produced by Raman scattering (RS).

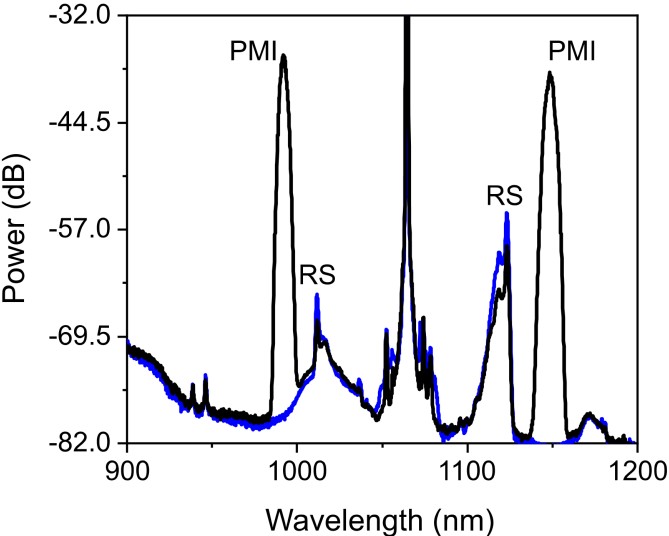

**Figure 9.** Light spectrum for two orthogonal polarization orientation of the pump. Pump polarization oriented to the fast axis (blue line) and to the slow axis (black line). Both spectra were recorded at pump peak power of 2.4 kW.

The polarization of the two strong bands shown in Figure 9 was experimentally analyzed with a bulk linear polarizer (LP). First, the HWP was adjusted to optimize the amplitude of the bands. The HWP was kept fixed during the experiment at this orientation. Then, the LP was inserted after the fiber end, and it was rotated to obtain best transmission of the bands. At this LP orientation, the amplitude of the PMI–SF bands was the maximum, while the amplitude of the rest of the spectral components (i.e., the residual pump and the RS bands) were attenuated a few tens of dB. When the LP was rotated 90° degrees from the earlier orientation, the amplitude of both PMI bands decreased into the noise level, while the rest of spectral components experienced an amplitude increase of a few tens of dB. This behavior confirms that the polarization state of PMI–SF bands is orthogonal to RS and the remaining pump light.

The generation of PMI–SF in the PCFs described in the Section 3 was investigated. Figure 10 shows the resulting optical spectra from each sample. Notice that high orders of PMI–SF were generated in some fibers. Table 4 summarizes the spectral position (and frequency shift) of first-order PMI–SF bands generated in the fibers.

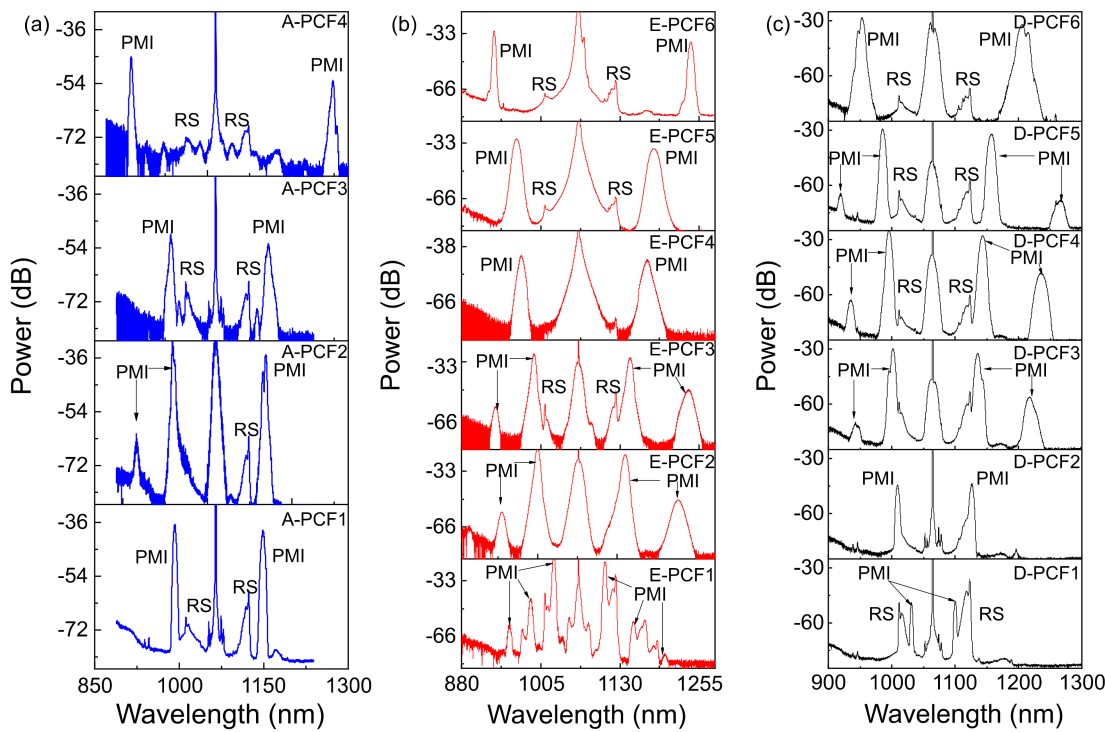

**Figure 10.** PMI spectra generated in (**a**) air-filled PCFs (blue line), (**b**) ethanol-filled PCFs (red line) and (**c**) $D_2O$-filled PCFs (black line).

**Table 4.** Wavelength and frequency shift of PMI–SF bands generated in air-filled, ethanol-filled and $D_2O$-filled PCFs. Modal birefringence.

| Filling Substance | Fiber | Anti-Stokes (nm) | Stokes (nm) | Frequency Shift (THz) | Birefringence |
|---|---|---|---|---|---|
| Air | A-PCF1 | 991.4 | 1148.3 | 20.57 | $1.64 \times 10^{-5}$ |
| | A-PCF2 | 988.0 | 1153.8 | 21.81 | $1.72 \times 10^{-5}$ |
| | A-PCF3 | 985.0 | 1158.2 | 22.80 | $9.56 \times 10^{-6}$ |
| | A-PCF4 | 914.6 | 1272.9 | 46.14 | $6.41 \times 10^{-5}$ |
| Ethanol | E-PCF1 | 1025.4 | 1106.0 | 10.57 | $6.28 \times 10^{-6}$ |
| | E-PCF2 | 1000.0 | 1138.2 | 18.25 | $1.03 \times 10^{-5}$ |
| | E-PCF3 | 993.7 | 1146.6 | 20.18 | $1.05 \times 10^{-5}$ |
| | E-PCF4 | 974.1 | 1173.7 | 26.22 | $1.60 \times 10^{-5}$ |
| | E-PCF5 | 965.2 | 1185.8 | 28.83 | $1.80 \times 10^{-5}$ |
| | E-PCF6 | 931.7 | 1242.0 | 40.28 | $2.24 \times 10^{-5}$ |
| $D_2O$ | D-PCF1 | 1031.0 | 1100.0 | 9.10 | $1.55 \times 10^{-6}$ |
| | D-PCF2 | 1009.2 | 1126.7 | 15.56 | $1.91 \times 10^{-6}$ |
| | D-PCF3 | 1002.0 | 1136.0 | 17.74 | $7.0 \times 10^{-7}$ |
| | D-PCF4 | 995.0 | 1144.0 | 19.58 | $3.40 \times 10^{-6}$ |
| | D-PCF5 | 985.6 | 1157.8 | 22.71 | $3.79 \times 10^{-6}$ |
| | D_PCF6 | 952.8 | 1206.1 | 33.09 | $5.21 \times 10^{-6}$ |

As it will be shown later in detail, in the case of ANDi PCFs with convex dispersion profile, the largest PMI shift was attained when the pump wavelength was close to the MDW, and the dispersion at the pump wavelength was as small as possible. This is shown experimentally in Figure 10, where the largest frequency shift (≈46 THz) occurs in fiber A-PCF4, with anti-Stokes and Stokes bands centered at 914 nm and 1273 nm, respectively. When the pump wavelength is far from the MDW, so that the fiber dispersion at the pump wavelength is larger, the frequency shift becomes smaller. As a representative example,

fiber D-PCF1 generated PMI–SF bands centered at 1031 and 1100 nm, which corresponds to a frequency shift of 9 THz, which was the smallest obtained in the experiments.

The frequency shift of PMI–SF generated in fibers with more conventional dispersion profiles can also be very large when the pump wavelength is close to a ZDW and the dispersion is small. From our experiments, we can investigate the generation of PMI in fibers with one ZDW with different chromatic dispersion at the pump wavelength. For example, the results obtained from the sequence of fibers E-PCF4, E-PCF5 and E-PCF6, with dispersion values decreasing from −13.7 ps/(nm·km) to −8.4 ps/(nm·km), show PMI frequency shift clearly increasing as the dispersion value decreases, from ≈26 THz in the case of E-PCF4 to ≈40 THz for E-PCF6.

A similar trend can be seen if we consider the series of fibers filled with $D_2O$, with the exception of D-PCF1 that exhibits an ANDi profile. The largest frequency shift (≈33 THz) occurred in fiber D-PCF6, which is the fiber with lowest dispersion, while the fiber D-PCF2 with largest dispersion showed the shortest frequency shift (≈15 THz)

It is worth noting that, in some fibers with similar values of dispersion at 1064 nm, PMI–SF produces bands with quite different frequency shifts. For example, the frequency shift in ANDi fibers A-PCF1 and D-PCF1 is ≈20 THz and ≈9 THz, respectively. Similarly, it happens for some fibers filled with $D_2O$. Residual phase birefringence is the origin of this apparent discrepancy. As stated in Equation (1), phase birefringence also plays a major role in the phase matching condition and adds positively to the frequency shift. The larger the birefringence, the more it contributes to the frequency shift.

The wavelengths at which the bands were generated varied accordingly with the dispersion characteristics and residual birefringence of each fiber. In general terms, experimental results can be described appropriately with theoretical calculations of PMI–SF wavelengths, taking into account the fibers' properties. Fiber birefringence values stated in Table 4 were obtained from fitting the theoretical calculations of frequency shift to the experimental results. The resulting values of birefringence range from $10^{-5}$ to $10^{-7}$, which are compatible with residual phase birefringence values in solid core PCFs. Figure 11 shows the theoretical results of PMI–SF frequency shift as function of pump wavelength for the different fibers, along with the experimental data.

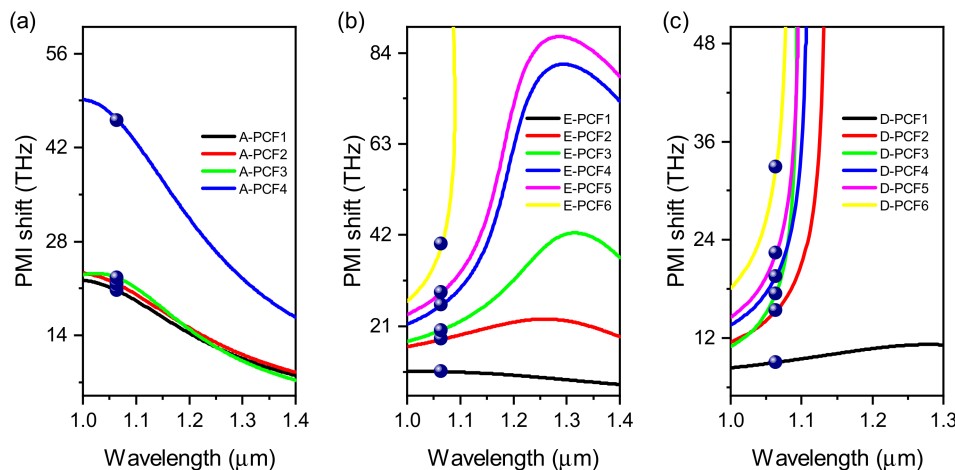

**Figure 11.** PMI–SF frequency shift as function of pump wavelength calculated for (**a**) air-filled PCFs, (**b**) ethanol-filled PCFs and (**c**) $D_2O$-filled PCFs. Dots are experimental data. Peak power of 2.5 kW for air-filled PCFs and 3 kW for both ethanol-filled PCFs and $D_2O$-filled PCFs.

As mentioned before, it is shown that the frequency shift in fibers with one ZDW shows a phase matching turning point at a wavelength close to the ZDW, so that PMI–SF can no longer occur for pump wavelengths above it. Large frequency shifts can be attained by pumping near the wavelength at which the turning point takes place, as it happens for fibers E-PCF6 and D-PCF6. Operating near the turning point might have also some

drawbacks: the large frequency shift slope is likely to cause the generation of broader bands. Additionally, the parametric wavelengths are very sensitive to small fluctuations of the fiber geometry, which, in practice, can lead to unwanted broadening of the generated bands.

In the case of PCFs with two relatively close ZDWs and ANDi PCFs, both showing convex dispersion profile, the largest frequency shift is attained when the pump wavelength is close to the MDW. Phase matching can occur regardless of the pump wavelength in the first type, even under anomalous dispersion pumping. However, it is important to remark that additional four-wave mixing processes, in particular, scalar FWM and modulation instability, can happen in fibers with dispersion profiles showing one or two ZDWs, while they are forbidden in ANDi fibers.

Finally, the strong dependence of PMI frequency shift on phase birefringence is shown in the theoretical simulations. For instance, comparing fibers A-PCF3 and A-PCF4, with rather similar dispersion characteristics, they show quite different PMI frequency shifts (see Figure 11a), due to the different contributions of birefringence to the phase mismatch.

## 6. Conclusions

In summary, we have reported a detailed study regarding PMI generation in a collection of PCFs. Air-filled, ethanol-filled and $D_2O$-filled PCFs featuring different chromatic dispersion characteristics have been investigated using long pump pulses at 1064 nm. It has been shown that PMI frequency shift can be very large in ANDi fibers with low dispersion values when they are pumped close to the MDW. The largest frequency shift observed in our experiments for ANDi fibers was 46 THZ in a PCF with $-13.1$ ps/nm·km of dispersion at the pump wavelength and phase birefringence of $6.4 \times 10^{-5}$. Large frequency shift is also observed in PCFs with dispersion profiles with one ZDW (or two ZDW) that were pumped near the ZDW. Frequency shifts of 40 THz and 32.9 THz were recorded for an ethanol-filled PCF (E-PCF6) and a $D_2O$-filled PCF (D-PCF6), respectively. In both cases, fibers were pumped near the PMI phase matching turning point. Phase matching condition and energy conservation were used to theoretically investigate the frequency shift of the PMI–SF process in the fibers investigated experimentally, showing good agreement with the experimental data.

In the present work, PMI has been investigated in a collection of fibers covering a wide range of structural parameters and dispersion regimes. We hope that the results reported here will be useful for future works that might involve optical fiber design for PMI generation.

**Author Contributions:** Conceptualization, A.D., A.L.-T. and M.V.A.; methodology, A.D. and A.L.-T.; software, E.S. and A.L.-T.; formal analysis, A.L.-T., A.D. and M.V.A.; investigation, A.L.-T. and A.D.; writing, A.L.-T., A.D. and M.V.A.; supervision, M.V.A.; project administration, M.V.A.; funding acquisition, M.V.A. and A.D. All authors have read and agreed to the published version of the manuscript.

**Funding:** Ministerio de Ciencia e Innovación and Fondo Europeo de Desarrollo Regional (PID2019-104276RB-I00); Generalitat Valenciana (PROMETEO/2019/048 and IDIFEDER/2020/064); and European Commission (H2020-MSCA-RISE-2019-872049).

**Conflicts of Interest:** The authors declare no conflict of interest.

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
