# Peer review of "Polarization Modulation Instability in Dispersion-Engineered Photonic Crystal Fibers"

_crystals, doi:10.3390/cryst11040365_

Round 1
Reviewer 1 Report
Follow editor

Author Response
Comment
- What is reason behind PMI spectral broadening with different filled materials.
Reply
The bandwidth of the PMI bands depends essentially on the dispersion properties of the optical fiber. For the specific fibers studied in this work, it just happens that the phase mismatch changes with frequency more quickly for the air filled fibers than for the rest of fibers, and consequently the gain spectrum for air filled fibers contains narrower bands.
Comment
- Did authors have also considered the effect of complex refractive index and negative nonlinear refractive index (n2). I can’t see the analysis of these two cases.
Reply
Theoretical simulations included in the paper are focused to describe the experimental observations. The study of optical nonlinear effects as PMI in a medium with absorption or a material showing negative n2 can indeed be very interesting, however we believe it is beyond the scope of this paper.
Comment
- Why there is higher frequency shift with higher stokes wavelength with all the filled material is fiber. Authors are suggested to established any mathematical relation for the same.
Reply
In the paper we stated the pump wavelength of the laser was 1064 nm. The exact wavelength of the pump laser is 1064.53 nm. The frequency shift values of table 4 have been updated after taking into account this correction.
Comment
- In figure 8, there are two mirrors used after HWP, the polarization state will change after 2nd mirror, what is the authors takes on that.
Reply
The polarization orientation of the pump beam at the fiber input is adjusted to match the polarization eigenstates of the fiber using the HWP. In the experiment, the HWP is rotated until the amplitude of the generated PMI bands is the highest. The polarization changes induced by the mirrors is compensated.
Comment
- Authors need to verify lectures in the manuscript.
Reply
The manuscript has been checked and revised thoroughly. Some small mistakes have been corrected.

Reviewer 2 Report
1.It's a clear presentation.
2.More clear results why choosing ethanol and heavy water are expected and the detailed material properties used in simulation are suggested as well.
3. The highest frequency shift occurs with the highest birefringence in Table 4. Does it help use elliptical shaped core (e.g. elliptical holes) to manipulate week birefringence maintaining?
5. Dispersion as a function of d/pitch and refractive index could help understand why choosing d/pitch of 0.36, 0.58 and 0.56 with filled material of air, ethanol and heavy water.
6. In conclusion, the guidelines might be pointed out clearly.
Author Response
Comment
- It's a clear presentation.
Comment
2. More clear results why choosing ethanol and heavy water are expected and the detailed material properties used in simulation are suggested as well.
Reply
The reasons why ethanol and heavy water were selected for filling the fibers are already stated in the manuscript. In section 2, line 130 it is written “For our experiments, we chose ethanol and heavy water due their suitable refractive index and low absorption in the infrared region.”
The refractive index of ethanol and heavy water used in the simulations were taken from reference [21]. In that reference, the dispersion of the refractive index with wavelength of both is reported as well. The following sentence has been added (line 132-133) to state it more clearly,
“Refractive index characteristics of both liquids used for the theoretical simulations are taken from [21]. Their wavelength dispersion was taken into account in the calculations.”
Comment
- The highest frequency shift occurs with the highest birefringence in Table4. Does it help use elliptical shaped core (e.g. elliptical holes) to manipulate week birefringence maintaining?
Reply
Generation of PMI is efficient in low birefringent fibers, regardless of the origin of such birefringence. Therefore, generation of PMI could in principle be optimized by engineering the fiber birefringence by any method, as for example the one suggested by the reviewer, as far as the birefringence remains small (10-5 or lower). PMI is due to a coherent coupling between orthogonal polarization states which have a small wave vector mismatch. The coherent coupling is assisted by a weak phase birefringence. In the propagation differential equations governing the evolution of the two polarization states along the fiber, there is a term that accounts for the coherent coupling between the two polarization components (that term leas to PMI generation). If the fiber length is much larger than the polarization beatlength, such term changes it sign many times along the propagation and its contribution averages to zero. Then, PMI is not produced at all.
Comment
- Dispersion as a function of d/pitch and refractive index could help understand why choosing d/pitch of 0.36, 0.58 and 0.56 with filled material of air, ethanol and heavy water.
Reply
Chapter 2 is included to give a theoretical background of the PMI process. The value of the parameters used in the simulations included in this chapter were chosen taking into consideration two aspects: (1) to show properly the different regimes for PMI that can occur depending on the fiber dispersion properties, and (2) to be similar to the experimental ones in order to be useful to back the experimental observations.
A full theoretical investigation of dispersion and PMI as a function of fiber geometry and holes refractive index could also be done, but we believe it is not the scope of this (mostly) experimental paper.
Comment
- In conclusion, the guidelines might be pointed out clearly.
Reply
Last paragraph of conclusions section has been rewritten.
